# Understanding Sparse Feature Updates in Deep Networks using Iterative Linearisation

## Abstract

Larger and deeper neural networks generalise well despite their increased capacity to overfit the data. Understanding why this happens is theoretically and practically important. A recent approach has investigated infinitely wide limits of neural networks through their corresponding *Neural Tangent Kernel*s (NTKs), demonstrating their equivalence to kernel regression with a fixed kernel derived from the network's architecture and initialisation. However, this "lazy training" cannot explain feature learning as such regimes correspond to linearised training in the neural network weight space, which implies a constant NTK kernel throughout training and, as such, does not perform feature learning. In practice, the empirical NTK kernel for finite networks can change substantially, particularly during the initial phase of stochastic gradient descent (SGD), highlighting the importance of feature learning. In this work, we derive *iterative linearisation* — an interpolation between SGD and the NTK kernel-based regression. Iterative linearisation enables us to precisely quantify the frequency of feature learning and is shown to be equivalent to NTK kernel-based regression in specific conditions. Empirically, only a surprisingly small amount of feature learning is required to achieve comparable performance to SGD, however, disabling feature learning negatively impacts generalisation. We further justify the validity of iterative linearisation by showing that with large periodicity, it is a special variant of the Truncated Gauss-Newton optimisation algorithm. We use this connection to provide novel insights on the role of damping on feature learning and generalisation in Truncated Gauss-Newton.

## 1 Introduction

Deep neural networks perform well on a wide variety of tasks despite their over-parameterisation and capacity to memorise random labels (Zhang et al., 2017). Even more surprisingly, generalisation behaviour improves as the number of parameters and capacity increases (Nakkiran et al., 2020). This correspondence between parameter count and generalisation goes contrary to classical beliefs around learning theory and overfitting, implying the existence of an inductive bias which encourages the networks to converge to well-generalising solutions. One approach to understanding this inductive bias has been to examine infinite width limits of neural networks using the *Neural Tangent Kernel* (NTK) (Jacot et al., 2018; Lee et al., 2019). In contrast to the performance scaling law for finite neural networks, infinitely wide neural networks often generalise worse than standard neural networks, despite being larger. However, under specific conditions, they can perform equivalently in certain scenarios (Lee et al., 2020). Similarly, despite their analytical tractability based on kernel regression, they do not accurately predict the behaviour of finite networks in many cases. For example, due to the lack of feature learning, they cannot be used for transfer learning (Yang & Hu, 2021). Additionally, the empirical NTK (outer product of Jacobians) changes significantly throughout training whereas NTK theory states in the infinite limit that this is constant. This finding raises important questions about the role of feature learning, driven by changes in the empirical NTK, in the generalisation behaviour of trained networks. Specifically, it prompts us to investigate how the frequency and extent of these feature updates, captured by the changing kernel, influence the network's ability to generalise to unseen data.

Following the NTK literature Lee et al. (2019), we define the network's features as the rows of the Jacobian matrix of the network's output with respect to its parameters. Note that this includes the final layer

activations as the derivative of the output of the network with respect to the final layer weights is the last layer activations (the features used for the Neural Network Gaussian Process Kernel). We also examine the first layer weights as features as a more intuitive definition and to show they align.

To address this challenge, we introduce iterative linearisation, a novel training algorithm that allows us to interpolate between the standard, feature-learning-rich training of neural networks and the "lazy training" (constant kernel) regime characteristic of infinite-width models under the NTK framework. Put simply, our method involves periodically freezing the network's features and training a simplified, linear model based on these frozen features. By adjusting the frequency of these "feature updates," we can control the extent to which the network learns new features. Our key insight is to control the frequency of feature updates during training. Instead of updating the features at every step as in standard SGD, we linearize the network around its current state and train this simplified, linear model for a fixed number of steps before updating the linearisation. This allows us to systematically study the impact of feature learning on generalisation. Through a series of experiments, we demonstrate that surprisingly infrequent feature updates are sufficient to achieve performance comparable to SGD, highlighting the efficiency of feature learning. Furthermore, we establish a connection between the feature update frequency in our method and damping in second-order optimisation algorithms, providing a new perspective on the role of damping in generalisation.

Iterative linearisation works by training a proxy linear model. At each step $s$, we approximate the network's behaviour around its current weights using a first-order Taylor expansion:

$$f_{s,t}^{\text{lin}}(x) = f_{\theta_s}(x) + \nabla_\theta f_{\theta_s}(x)^\top (\theta_t - \theta_s) \tag{1}$$

The proxy model (Equation (1)) is then trained for $K$ steps before we re-linearize, effectively taking a new snapshot of the network's features to train a proxy linear model on. Training a linear proxy keeps the Jacobian constant in between, thus the empirical NTK ($\nabla_\theta f_{\theta_s}(x) \nabla_\theta f_{\theta_s}(x)^\top$) also stays constant, implying that no feature learning is happening and following the NTK theory for those $K$ steps.

In parallel, we show that iterative linearisation with large $K$ approximates the Truncated Gauss-Newton (TGN) algorithm and use this to connect our analysis to second order methods (Section 3.2). We examine damping in TGN and connect it to the parameter $K$ in iterative linearisation, finding that damping in TGN has a similar effect on performance as reducing $K$, allowing more feature learning and better generalisation.

Contributions

- We formalise a new training algorithm, *iterative linearisation*, that allows us to control feature learning frequency and interpolate between lazy (NTK) and rich (feature learning) regimes of training, providing a powerful tool for investigating the role of feature learning in neural networks
- We show empirically that only a small amount of feature learning is required for comparable generalisation performance to SGD. Examples are also provided where too few feature updates results in worse generalisation performance when train loss converges before features can be learnt.
- We investigate *iterative linearisation* with a low learning rate and large period and show intrinsic connections to second order methods. We also provide intuition and empirical evidence as to how damping in second order methods is connected to more regular feature updates.
  We note that iteratise linearisation is a *tool for understanding feature learning and generalisation in neural networks*. We do not propose it as a replacement for SGD as it is computationally expensive. As such our experiments are limited to smaller models and two domains (simple low dimensional datasets and image classification).

## 2 Background

### 2.1 Notation

Throughout this section and the rest of the paper we use the following notation:

- Parameters: $\theta$, parameters at time step $t$: $\theta_t$

- Dataset inputs & labels: $X, Y$
- Neural network with parameters $\theta$: $f_\theta(\cdot)$
- Neural network Jacobian of network outputs with respect to weights at time $t$: $\phi_t = \nabla_\theta f_{\theta_t}(X)$
- Neural network linearised at parameters $\theta_s$ and evaluated with parameters $\theta_t$: $f_{s,t}^{\mathrm{lin}}(x) = f_{\theta_s}(x) + \phi_s^\top(\theta_t - \theta_s)$

## 2.2 Lack of feature learning in the NTK regime

In the infinite-width limit, neural networks exhibit an intriguing phenomenon. Their training dynamics can be described by a fixed kernel known as the Neural Tangent Kernel (NTK) (Jacot et al., 2018). This kernel, which depends only on the network architecture and initialisation, remains constant throughout training. This "lazy training" (constant kernel) regime implies that the network behaves like a linear model and, crucially, no feature learning occurs. While theoretically appealing, this regime often fails to capture the generalisation behaviour of finite-width networks used in practice, where feature learning plays a vital role. We begin with a brief summary of the NTK results and refer the reader to Lee et al. (2019) for more details.

First, let us consider a neural network $f_\theta(X_i)$, where the network parameters $\theta$ are iteratively updated through Gradient Descent.

$$\theta_{t+1} = \theta_t - \eta\nabla_\theta\mathcal{L}(f_{\theta_t}(X), Y) \tag{2}$$

With learning rate $\eta$ and loss $\mathcal{L}$. We consider the MSE loss $\mathcal{L}(\hat{Y}) = \frac{1}{2}\|\hat{Y} - Y\|_2^2$[1]. We use $X$ for the training inputs and $Y$ for the labels.

Moving to the continuous-time limit, we have *Gradient Flow*, where the learning rate approaches 0 and parameter updates become a continuous trajectory (we use $\phi_t$ for the Jacobian at time $t$):

$$\dot{\theta}_t = -\nabla_\theta f_{\theta_t}(X)^\top(f_{\theta_t}(X) - Y) = -\phi_t^\top(f_{\theta_t}(X) - Y) \tag{3}$$

Note that Equation (3) shows the gradient flow equation specifically for the MSE loss, resulting in the residual term $(f_{\theta_t}(X) - Y)$.

We now look at how the function $f_{\theta_t}(X_i)$ itself changes. We apply the chain rule $\dot{f}_{\theta_t}(X) = \frac{\partial f_{\theta_t}(X)}{\partial \theta}\dot{\theta}_t = \phi_t\dot{\theta}_t$, substituting in Equation (3).

$$\dot{f}_{\theta_t}(X) = -\underbrace{\left[\phi_t\,\phi_t^\top\right]}_{\text{empirical NTK}}(f_{\theta_t}(X) - Y). \tag{4}$$

The term in brackets is the *empirical Neural Tangent Kernel* (empirical NTK), $\hat{\Theta}_t(X, X)$ where the $i,j$th entry is the inner product of the gradients of the network with respect to the parameters for datapoints $i$ and $j$:

$$\left[\hat{\Theta}_t(X, X)\right]_{ij} = \left\langle\frac{\partial f_{\theta_t}(X_i)}{\partial\theta}, \frac{\partial f_{\theta_t}(X_j)}{\partial\theta}\right\rangle = \left[\phi_t\,\phi_t^\top\right]_{ij} \tag{5}$$

Thus, function evolution is:

$$\dot{f}_{\theta_t}(X) = -\hat{\Theta}_t(X, X)(f_{\theta_t}(X) - Y) \tag{6}$$

In the infinite width limit the empirical NTK, $\hat{\Theta}_t$, converges to a constant kernel, $\Theta$, which we call the *Neural Tangent Kernel* (NTK) (Jacot et al., 2018; Lee et al., 2019; Arora et al., 2019). This is a matrix dependent only on the architecture and initialisation scheme, which does not change throughout training. From this perspective, training the model in this infinite-width limit under gradient flow (or gradient descent with a

---

[1]We use MSE for simplicity here. While this is needed for some NTK results, it does not affect the algorithms we propose where any differentiable loss function can be used — see Appendix A

small step size) is equivalent to training the weight-space linearisation of the neural network (Lee et al., 2019). Hence as $\Theta$ is constant, the network evolves as a linear model and the features do not change.

In contrast to the infinite width case, linearising finite networks results in significantly worse test loss (Lee et al., 2020). One common explanation is that this is due to the lack of sufficient random features at initialisation, highlighting the importance of feature learning. In comparison, running gradient descent on the full network allows features to be learned, thus reducing the reliance on initial random features. This highlights the crucial role of feature learning in the success of finite-width neural networks. Note that results on pruning at initialisation (Ramanujan et al., 2020) in place of training are distinct from fixing the features in iterative linearisation as pruning causes hidden layer representations to change and hence features are affected by pruning.

### 2.3 Gauss-Newton algorithms

We want to use a correspondence to the Gauss-Newton algorithm to provide insights into our proposed algorithm. In this section we define the Gauss-Newton algorithm and Truncated Gauss-Newton (TGN, from Gratton et al. (2007)), along with how they are related in preparation for this.

---

**Algorithm 1** The Gauss-Newton algorithm

---

**Input:** initial parameters $\theta_0$, data $\{X, Y\}$
  $t \leftarrow 0$
  **while** not converged **do**
    $\phi_t \leftarrow \nabla_\theta f_{\theta_t}(X)$
    Find an $s_t$ s.t. $(\phi_t^\top \phi_t)s_t = -\phi_t^\top (f_{\theta_t}(X) - Y)$
    $\theta_{t+1} \leftarrow \theta_t + s_t$
    $t \leftarrow t + 1$
  **end while**

---

**Algorithm 2** Truncated Gauss-Newton (TGN)

---

**Input:** initial parameters $\theta_0$, tolerances $\beta_t$, data $\{X, Y\}$
  $t \leftarrow 0$
  **while** not converged **do**
    $\phi_t \leftarrow \nabla_\theta f_{\theta_t}(X)$
    Find an $s_t$ s.t. $(\phi_t^\top \phi_t)s_t = -\phi_t^\top (f_{\theta_t}(X) - Y) + r_t$    with $\|r_t\|_2 \leq \beta_t \|\phi_t^\top (f_{\theta_t}(X) - Y)\|_2$
    $\theta_{t+1} \leftarrow \theta_t + s_t$
    $t \leftarrow t + 1$
  **end while**

---

The Gauss-Newton algorithm is a variant of Newton's Method where the Hessian $H$ is replaced with an approximation based only off of the first-order gradients $\hat{H} = \phi_t^\top \phi_t$. Here $\phi_t$ are the gradients of the network with respect to parameters at time $t$. Where Newton's method takes a step of $\Delta\theta = -H^{-1}\phi_t^\top (f_{\theta_t}(X) - Y)$, Gauss-Newton instead takes a step of $\Delta\theta = -(\phi_t^\top \phi_t)^{-1} \phi_t^\top (f_{\theta_t}(X) - Y)$. We write it here assuming mean squared error as our loss function (causing the residuals to appear in the gradient above) but the *Generalised* Gauss-Newton (Schraudolph, 2002) algorithm extends this. Iterating this step results in the Gauss-Newton algorithm and approximates Newton's method. It can equivalently be viewed as first linearising the model and then solving exactly for the Newton step. For example, minimising the least squares loss $\mathcal{L}(\theta) = \|f_\theta^{\text{lin}}(X) - Y\|_2^2$ on the first order Taylor expansion $f_\theta^{\text{lin}}(x) = f_{\theta_t}(x) + \phi_t^\top (\theta - \theta_t)$ results in the same update step.

Damping in the form of $\lambda I$ being added before inverting or an explicit step size are common variations to improve numerical stability and optimisation convergence.

Truncated Gauss-Newton (TGN) relaxes the exact linear solve by allowing an error term $r_t$ such that the linear least squares problem is only solved within a tolerance of $r_t$. Such a solution could occur if we solve the

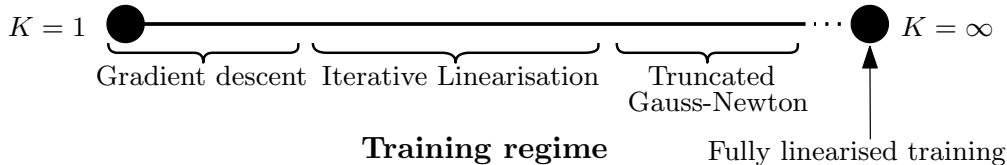

Figure 1: Training regimes when always training to convergence. For small values of $K$, it will follow a similar training trajectory to gradient descent. With a low enough learning rate, for very large $K$, it will train until convergence for each linearisation giving the same results for all larger $K$ — this is equivalent to the Truncated Gauss-Newton algorithm. When $K = \infty$ then fully linearised training is performed. In the middle is a section we label *iterative linearisation* which does not fit cleanly into any of the other regimes.

linear model using SGD rather than inverting the matrix and do not achieve exact convergence. TGN is a useful algorithm for understanding the effect of not solving the linearised problem exactly and was intended to understand the effect of terminating slightly before convergence or having numerical precision issues. Gratton et al. (2007) (Section 5.2) proves that under moderate conditions (twice continuously differentiable, existence of a critical point such that the Jacobian has full rank and $\beta_t$ sufficiently small) TGN maintains the local convergence of Gauss-Newton.

## 3  Iterative Linearisation

NTK theory says that if the width is large enough, training the weight-space linearisation is equivalent to training the full network (Lee et al., 2019). However, in practice, training the fully linearised network performs very poorly for practically sized networks (Lee et al., 2020). In this section, we formalise *iterative linearisation* in order to interpolate between the training of the standard network and the linearised network.

Consider standard (full batch) gradient descent on a neural network with a squared error loss (Equation (7)).

$$\theta_{t+1} = \theta_t - \eta \phi_t^\top (f_{\theta_t}(X) - Y) \quad \text{where} \quad \phi_t = \nabla_\theta f_{\theta_t}(X) \tag{7}$$

Here we can view the weights, $\theta_t$, and features, $\phi_t$, as two separate variables we update each step. However, there is no requirement that we always update both, giving rise to the following generalised algorithm (Algorithm 3) where we only update the features every $K$ steps:

$$\theta_{t+1} = \theta_t - \eta \phi_s^\top (\ \underbrace{f_{\theta_s}(X) + \phi_s^\top (\theta_t - \theta_s)}_{\text{Linearisation of } f_{\theta_t}(X) \text{ at } \theta_s} \ -Y) \tag{8}$$

$$\phi_s = \nabla_\theta f_{\theta_s}(X) \tag{9}$$

where $s = K * \lfloor \frac{t}{K} \rfloor$ such that every $K$ steps, the neural network $f(\cdot)$ is *re-linearised* through its first order Taylor expansion at the weights $\theta_s$. We note that except at these re-linearisation steps, the features being used do not change. In the extreme case of $K = \infty$, the NTK training dynamics are followed and a linear model which never learns features is trained.

Algorithm 3 shows pedagogically how this would work. While our original formulation used MSE loss, any differentiable loss function can be used. For instance, with softmax cross-entropy, we linearise the network but keep the softmax and cross-entropy in the (non-linearised) loss function. This ensures that the output remains a probability distribution and prevents numerical issues. To avoid the need to store the Jacobian $\phi_t$ explicitly, we perform a first-order Taylor expansion which can be computed efficiently using Jacobian-vector products. This also enables stochastic updates via SGD rather than full-batch gradient descent, improving scalability.

Using this framework, when $K = 1$ this is simply gradient descent and when $K = \infty$ it is fully linearised training. Other values of $K$ interpolate between these two extremes but for all $K < \infty$ it will do at least some feature updates. See Figure 1 for more details. Note that we can also generalise this to be nonperiodic in terms of when we update $\phi$ so we call this *fixed period* iterative linearisation.

---

**Algorithm 3** Iterative Linearisation (fixed period)

---

**Input:** learning rate $\eta$, update periodicity $K$, initial parameters $\theta_0$, data $\{X, Y\}$

   $t \leftarrow 0$
   **while** not converged **do**
      $\phi_s \leftarrow \nabla_\theta f_{\theta_t}(x)$ **if** $t \bmod K = 0$ **else** $\phi_{t-1}$
      $\theta_{t+1} \leftarrow \theta_t - \eta \phi_s^\top (\ \underbrace{f_{\theta_s}(X) + \phi_s^\top(\theta_t - \theta_s)}_{\text{Linearisation of } f_{\theta_t}(X) \text{ at } \theta_s}\ - Y)$
      $t \leftarrow t + 1$
   **end while**

---

### 3.1 Control of feature learning frequency

Consider Algorithm 3, if $K = \infty$ then training only involves Equation (8), this is purely linearised training using random features defined by the neural network's initialisation function. At this step, no feature learning takes place. This is similar to linear models with fixed features. From this interpretation, the entries of Jacobian $\phi_t$ are the features we use at time $t$, and $K$ determines how frequently we update these features. As we decrease K, we introduce more frequent feature updates, gradually moving away from the NTK regime and towards the behavior of SGD. Thus, K serves as a dial that allows us to smoothly interpolate between these two extremes.

We point out that feature learning only happens in Equation (9), not in Equation (8). This inspires us to call Equation (9) the *feature learning* step. Furthermore, we conjecture the amount of feature learning from training the proxy model (linearised NN) for $K$ steps to be less than that from training the true model (unlinearised NN) for $K$ steps. See Section 4.2 for empirical evidence for this fact. Intuitively, this will be true due to the proxy model losing the learning signal information of the neural network. Therefore, after training the linearised model for some amount of steps, any changes to the features are due to randomness.

### 3.2 Validity and Convergence of Iterative Linearisation

Iterative linearisation is clearly reasonable for $K = 1$ as this reduces to standard SGD. For small $K$ we would expect similar behaviour (we show this empirically later in Section 4.1). It is not immediately obvious that iterative linearisation with large $K$ constitutes a valid learning algorithm. In this section we show a fundamental connection between large $K$ iterative linearisation and Gauss-Newton/Truncated Gauss-Newton in order to show why iterative linearisation still has local convergence with large $K$.

To establish the validity of iterative linearisation as a learning algorithm, particularly for large values of K, we first demonstrate a fundamental connection to the well-established Gauss-Newton method. In Gauss-Newton (Algorithm 1), each iteration requires finding any $s_t$ that satisfies the equation. This is typically found in small instances by inverting the Hessian approximation $(\phi^\top \phi)$, but $s_t$ can be computed using gradient descent instead, as is often done in practice for larger models. Note that this is an approximation to true Gauss-Newton, however under certain convergence tolerances this approximation (Truncated Gauss-Newton) also maintains local convergence Gratton et al. (2007).

Assuming the learning rate is chosen such that gradient descent will converge, then it will reach a point where it is within some tolerance of the true solution in a finite time. This is where the analysis of TGN comes in as it provides local convergence guarantees under certain conditions of the tolerances, (Gratton et al., 2007).

To summarise, if we were to solve our linear model exactly, then we would recover the Gauss-Newton algorithm. In practise no matter how large $K$ is, it won't converge perfectly, however to show local convergence building off of Gratton et al. (2007), we only need approximate convergence within some tolerance. If the learning rate is small enough then this is guaranteed for large $K$.

We now turn our attention to the more general scenario where the linear subproblem is solved approximately. In contrast to TGN, iterative linearisation allocates a fixed computational budget of K steps to solve the

linearised subproblem, regardless of its proximity to the true solution. This lets us explore the areas where it is not equivalent to Truncated Gauss-Newton which are interesting in the context of feature learning in neural networks, however it means that we need the extra step below to show local convergence.

The preceding arguments suggest that iterative linearisation should converge locally for sufficiently large K. To formalize this, we now leverage the local convergence properties of Truncated Gauss-Newton (TGN). We can see from the local convergence of TGN that there must exist a $K$ such that iterative linearisation also has local convergence. We do this by using a low enough learning rate to achieve convergence in the linear least squares problem and setting $K$ large enough to have $\|r_t\|_2 \leq \beta_t \|\phi_t^\top (f_{\theta_t}(X) - Y)\|_2$ on each iteration with appropriately small $\beta_t$ as given by Theorem 5 in Gratton et al. (2007). Therefore, for sufficiently large $K$, iterative linearisation exhibits local convergence under the same conditions that guarantee the convergence of TGN.

If finding $s_k$ with the inverse in the overparameterised setting, then in general *damping* will be required to ensure that the matrix is invertible. This is due to the existence of many solutions to $(\phi_t^\top \phi_t)s_t = -\phi_t^\top (f_{\theta_t}(X) - Y)$ in the overparameterised regime. Under enough damping such that the solutions at each step are unique, all methods of finding $s_k$ will be identical up to numerical precision. In an overparameterised setting with no damping this is not as clearly true. We provide an intuition below that there exist periodicity $K$ and learning rate $\eta$ such that even in this setting it will find the same solution for each step if it can be solved by inverting the matrix. We begin with a lemma on the closed-form solution of gradient flow (this is a known result, but we include a proof in Appendix B for completeness).

**Lemma 3.2.1.** *Gradient flow on a linear model $f_\theta(x) = \phi(x)^\top \theta$ with squared error loss $\mathcal{L}(Y) = \frac{1}{2}\|Y - \hat{Y}\|_2^2$ starting at weights $\theta_0$ converges to $\theta_0 - (\phi^\top \phi)^{-1}\phi^\top r\left(f_{\theta_0}(X)\right)$, when $\phi^\top \phi$ is invertible. Where $\phi = \nabla_\theta f_{\theta_0}(X)$ is the Jacobian on the dataset $X$ and $r(Y) = (Y - \hat{Y})$ is the loss residual.*

Assume the same setup as before with a squared error loss. Additionally assume that $(\phi_t^\top \phi_t)$ is invertible at re-linearisation $t$. Solving a linear least squares problem using gradient descent, with a small enough learning rate will converge to the same solution as gradient flow. Lemma 3.2.1 shows that gradient flow will converge to $\theta_0 - (\phi_t^\top \phi_t)^{-1}\phi_t^\top \nabla_\theta(f_{\theta_t}(X) - Y)$. Hence for each linearisation there exists a learning rate such that for smaller learning rates, it will converge to the solution obtained by inverting the matrix. We set $K$ to be large enough to converge to within numerical precision. Taking the minimum learning rate across all linearisations and the maximum time horizon needed for any of those to converge allows a global setting of $\eta$ and $K$ across all linearisations.

This exact equivalence relies on a squared error loss, however we can easily generalise to the idea of exactly solving the convex problem that results from any convex loss function on the linearised neural network. Though there will no longer be a closed-form solution, we can still solve this numerically as we will do in Section 4.3. From this perspective, we can see the Gauss-Newton algorithm as the special case where we exactly solve iterative linearisation on a squared error loss.

### 3.3 Connecting Damping with Feature Learning: Insights from Gauss-Newton

As shown above, iterative linearisation approaches the Truncated Gauss-Newton algorithm as $K$ increases if the learning rate is small enough. This connection is not merely a mathematical curiosity; it provides crucial insights into the interplay between optimisation dynamics, feature learning, and generalisation. We now delve deeper into this connection, examining iterative linearisation from the perspective of damping in second-order methods to understand how it affects feature learning.

To motivate the connection to iterative linearisation, we first briefly revisit the conventional role of damping in second-order optimisation methods like Gauss-Newton. When using second order methods in practise, *damping* is used in order to help with numerical stability for inverting the matrix and provide a trust region for the proxy model to prevent excessively large steps that could lead to divergence or poor convergence. Damping is typically implemented by adding a scaled identity matrix to the Hessian approximation before inversion. Beyond its primary purpose, we show that damping has an interesting, and perhaps unexpected, connection to the frequency of feature updates in iterative linearisation.

Adding damping to Gauss-Newton gives the new update below in Equation (10).

$$\theta_{t+1} = \theta_t - (\phi_t^\top \phi_t + \lambda I)^{-1} \phi_t^\top (f_{\theta_t}(X) - Y) \tag{10}$$

While the solution to the linearised network with squared error loss $\mathcal{L}(\theta) = \|f_{\theta_0}^{\text{lin}}(\theta; X) - Y\|^2$ is given by the Gauss-Newton step, the solution to the linearised network with loss $\mathcal{L}(\theta) = \|f_{\theta_0}^{\text{lin}}(\theta; X) - Y\|^2 + \lambda\|\theta - \theta_0\|^2$ is given by Equation (10).

We use this formulation of Gauss-Newton to highlight a correspondence between more damping and smaller $K$ in iterative linearisation and hence with more feature learning. The two methods converge as $\lambda \to 0$ and $K \to \infty$, becoming the Truncated Gauss-Newton algorithm. As $\lambda \to \infty$, damped Gauss-Newton approaches gradient flow, while as $K \to 1$, iterative linearisation approaches gradient descent. However, for a small enough learning rate, these are equivalent. Intuitively, as $\lambda$ increases, the regularisation term increasingly penalises large changes in the parameters, causing the linear model to be solved less precisely within the $K$ steps. Similarly, with smaller $K$, there are only a limited number of gradient steps; hence the network parameters cannot move as far, and the linearised proxy model is solved less completely. A similar argument can be applied to damping through an explicit step size for the Gauss-Newton step instead of a multiple of the identity added to the Gauss-Newton matrix.

Intuitively, damping restricts the optimisation process from making drastic changes to the weights in a single step. This can be seen as analogous to limiting the frequency of feature updates in iterative linearisation. In both cases, the model is encouraged to make smaller, more incremental adjustments, which, as we will show empirically, can lead to better generalisation.

This effect of damping is, to our knowledge, beyond what is understood in the literature. Generally, damping is purely used for numerical stability and trust region effects, rather than generalisation or feature learning.

Such insights raise an interesting hypothesis that the benefit of second-order methods requiring fewer steps may, at the extreme, result in worse generalisation in neural networks. This would be due to the lack of feature updates before convergence and provides interesting insight into why damping may be important beyond the numerical stability arguments normally used to justify it. We provide some evidence of this hypothesis in Section 4.3.

## 4 Experiments

We run experiments with both a simple CNN (a modified LeNet with extra channels – see Appendix C.2.1) and a ResNet18 on CIFAR10 to understand the effect of changing feature learning frequency. We point out that we do not aim for state-of-the-art performance (the networks only get to maximum $\sim 80\%$ for CIFAR10) as it is not necessary to prove our claims. The key comparisons are between large and small values of $K$ in iterative linearisation, and between various damping values in second order methods. We use cross-entropy loss in these experiments as, unlike NTK theory, no part of our derivation relied on MSE. In order to improve numerical stability and ensure that the output is a probability distribution during training, we do not linearise the softmax function. We also run some experiments in simple 1 and 2-dimensional problems to further examine damping in Gauss-Newton and visualise feature learning. Further experiments, model definitions and experimental details can be found in Appendix C.2.

We split out the experiments to support each of our core claims.

- Performance on par with SGD can be obtained with surprisingly few feature updates. Section 4.1 shows that with very few feature updates we can achieve similar test accuracy to SGD.

- Feature update frequency provides a good proxy for feature learning. Section 4.2 shows that large increases in $K$ reduces the total amount of feature learning before convergence in a number of settings and definitions of feature learning.

- Connecting to second order methods. Section 4.3 shows the impact of damping on performance and relates that back to feature learning and the periodicity ($K$) in iterative linearisation.

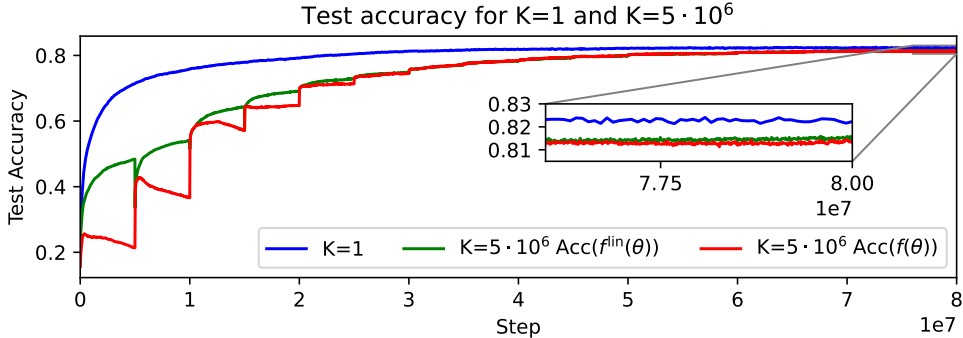

Figure 2: Standard SGD and data augmentation (flips and random crops) for large $K$ on a standard CNN architecture. This shows test accuracy for $K = 1$ as well as both the neural network test accuracy and the test accuracy of the linearised proxy network being trained. As can be seen, the performance is almost equivalent for very large $K$ given enough time to train. Note that as re-linearisation can occur in the middle of an epoch but test performance is evaluated at the end of an epoch, the lines do not always drop to the same point, this is simply an artefact of the measurement rather than underlying differences.

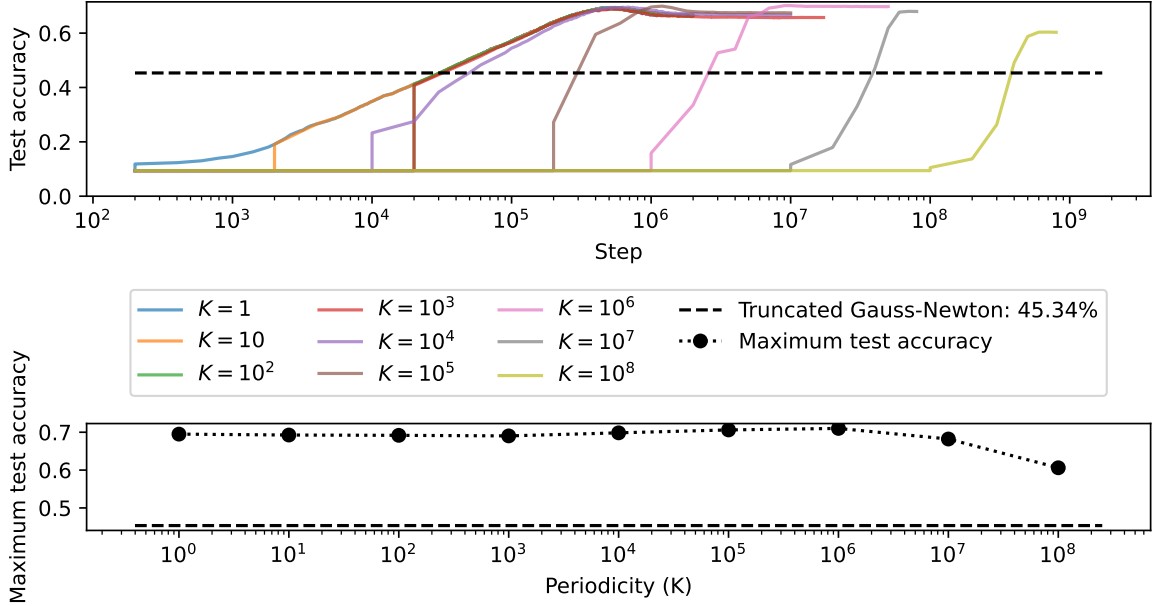

Figure 3: A sweep of $K$ for a basic CNN, always training until convergence. Up until $K = 10^4$, it follows the same trajectory, however after that it briefly improves slightly before getting significantly worse generalisation performance at $K = 10^8$. We include a baseline of running Truncated Gauss-Newton on this plot for comparison. We note that the first reading (except for at step 0) is at $\max(100, K)$ and plot the accuracy at initialisation until then.

## 4.1 SGD performance can be achieved with a surprisingly small number of feature updates

We first compare large $K$ to $K = 1$ to show that only infrequent feature updates are needed in order to achieve comparable generalisation performance for this dataset. We show this in a typical setting of a CNN on CIFAR10 adding data augmentation (Figure 2) where the final results differ by under 1% (82.3% for

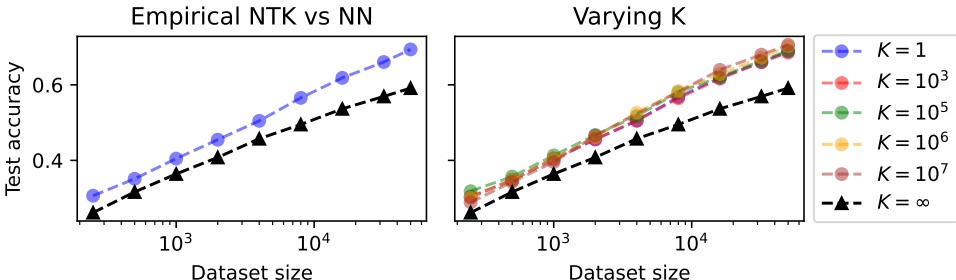

Figure 4: Data scaling behaviour of models as $K$ changes. Here, we show the performance of neural networks with various $K$ when trained on subsets of the training data and trained until convergence. The rate at which performance improves as the dataset size increases is equivalent for all finite $K$ and worse for the case where $K = \infty$ and the features are never updated. We note that $K = 10^8$ was too computationally expensive to run for each of these but does begin to do worse than other values of $K$ here. Note that the $K = \infty$ runs used a higher learning rate ($\eta$=1e-3 instead of $\eta$=1e-5) for computational reasons, so we include on the left plot a comparison with $K = 1$ with the same higher learning rate to show that this is not relevant to the comparison.

$K = 1$ and 81.4% for $K = 5 \times 10^6$). This result also holds when using ResNet18, as can be seen in Figure 10. With BatchNorm it is far more sensitive due to linearisation of the normalisation layers not playing well with batch statistics. It is extremely clear across these results that we get enough feature learning in 8-12 feature updates to achieve comparable generalisation to millions of feature updates when $K = 1$. As shown in Figure 2, we achieve comparable generalisation (within 1%) while performing over 100,000 times fewer feature updates.

In Figure 3, we vary $K$ over 8 orders of magnitude for a basic CNN on CIFAR10 to understand the full spectrum of the interpolation. Note that the first reading after step 0 (not on a log graph) is step $\max(100, K)$, in order to show the initial performance we plot it as that performance up until the first reading. Up until $K = 10^4$, it performs about the same as $K = 1$ but requiring $K$ times fewer feature updates, though a similar number of steps. After $K = 10^4$, performance begins to be erratic with it converging to a slightly higher performance for a bit before dropping down at $K = 10^8$. This convincingly shows that there is a wide range of $K$ values ($1 \leq K \leq 10^4$) which are approximately the same as $K = 1$. At $K = 10^8$, we begin to approach Truncated Gauss-Newton as the performance drops. The orders of magnitude in the middle are where Figures 2 and 10 both fall, which tend to perform similarly well but can be unstable. We note that our Truncated Gauss-Newton baseline is computed by solving the convex loss of the linearised model using Adam as we do later in Section 4.3.

Previous graphs only compare finite $K$. To address this, we reproduce and extend the approach used in Vyas et al. (2022) in Figure 4 and show that using the empirical kernel with $K = \infty$ scales poorly with dataset size. Thus, a complete lack of feature learning hurts generalisation. However, any finite $K$ chosen scales similarly to $K = 1$ if trained until convergence on the training data. This is because $K$ is *too small* to make a difference in whether the amount of feature learning needed for the task is achieved. Specifically, by the time it converges to 100% train accuracy, it has already learned enough features. We do not run for $K = 10^8$ here for computational reasons but we note that larger values of $K$ when we train do begin to reduce performance as we approach Truncated Gauss-Newton. As such, for the setting we have, it is clear that feature learning improves sample complexity however it is also clear that not much feature learning is necessary to achieve comparable sample complexity.

Overall we have shown conclusively that for this setting, only a few feature updates are needed for comparable performance. However there is still often some small generalisation benefit of the more feature learning that occurs with lower $K$.

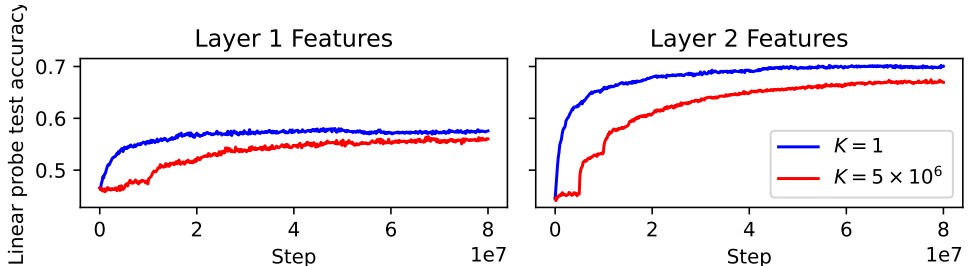

Figure 5: Test accuracy of a linear probe trained on the features of the first (left) and second (right) layers of features. As training progresses, the linear separability of the feature representations learnt at these layers improves, with the large $K$ features improving slower and levelling off at a lower point.

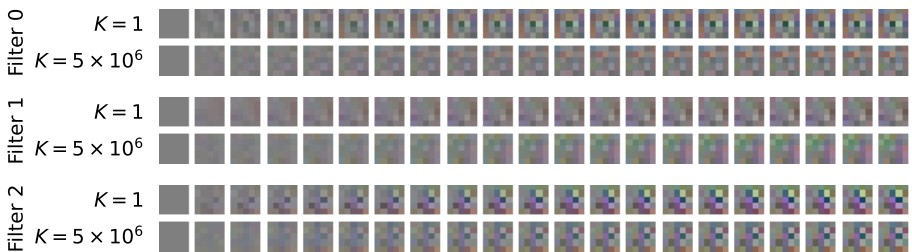

Figure 6: Evolution of the first three convolutional filters of the first layer. Each column represents 10000 epochs and for each filter there is the evolution for $K = 1$ and $K = 5 \times 10^6$. Each image maps the [-0.5, 0.5] range of the filter difference from initialisation to [0,1] in order to be plotted. As can be seen, the evolution of the filters for $K = 1$ is much faster and the final filter for $K = 5 \times 10^6$ is often similar to $K = 1$ at about 20% of the way through training.

### 4.2 Feature update frequency provides a good proxy for the amount of feature learning

One of the primary uses of iterative linearisation is to examine the effects of changing the amount of feature learning. In order to do this, we first show that the amount of feature learning decreases with increased periodicity. As feature learning is a complex process without a single quantitative measure, we use three distinct methods to understand its behaviour: the linear separability of internal representations (linear probes), direct visualization of convolutional filters, and analysis of learned weights on a synthetic task. If increasing $K$ is truly reducing feature learning, we would expect to see a corresponding degradation in the quality of the learned features, potentially leading to worse generalisation. To investigate this, we employ two distinct methods to analyze the features learned during training for the model shown in Figure 2: linear probes and visualisation of convolutional filters. We additionally show the first layer features in an XOR toy problem where they can be plotted in 2 dimensions.

For small $K$, it is quite clear from Figure 3 that the network learns more per feature update, requiring fewer feature updates to achieve the same test accuracy. This raises the question of when iterative linearisation results in less overall feature learning versus simply clumping the feature learning from $K$ steps into 1.

We analyse the features which are learnt during the training of Figure 2 in two ways. In Figure 5, we use linear probes (Alain & Bengio, 2017) in order to plot the linear separability of the learnt features for the task. This was done through training linear classifiers on the first and second-layer features and plotting the linear probe test accuracy. We can see that the features for $K = 5 \times 10^6$ improve slower than for $K = 1$ and levels off at a lower generalisation accuracy. This graph gives evidence that increasing $K$ also reduces the amount of feature learning, not only the frequency. However, this relationship is not linear: a single $K = 1$ results in less overall feature learning than $10^5$ steps with $K = 10^5$. The 1% performance gap in Figure 2 is likely partially due to the smaller amount of feature learning shown here. However, considering

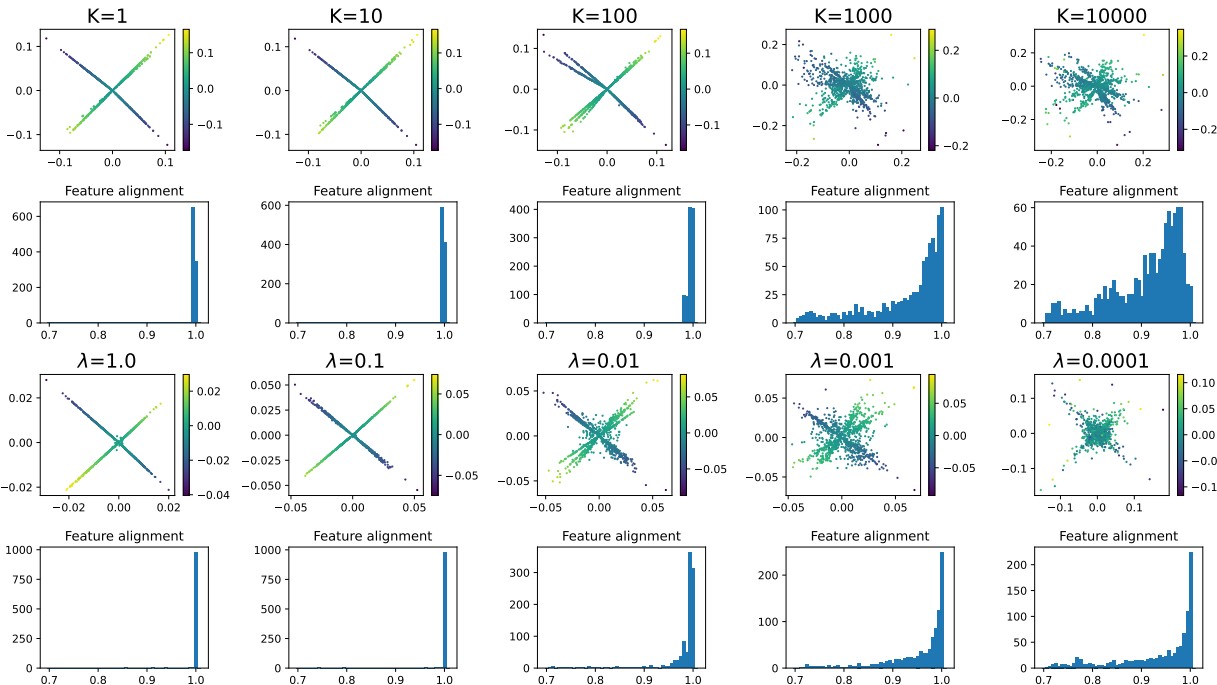

Figure 7: Converged features after training a single hidden layer model $\mathbf{w}_2\sigma(\mathbf{w}_1^\top\mathbf{x}+\mathbf{b})$ on XOR data for various values of $K$ and $\lambda$. The feature location is the first layer weight rows ($\mathbf{w}_{1i}$ for all $i$) and is coloured by the corresponding second layer weight $\mathbf{w}_{2i}$ . The second row shows the *feature alignment* which is defined by the histogram of maximal cosine similarity with first layer features and any of the four true cluster means, see Equation (11) for definition. The bottom two rows are similar but for different values of $K$. We see a similar change though the specifics are different as $K$ and $\lambda$ are not exactly comparable.

the relative gap in linear probe accuracy versus the gap in generalisation, it is clear that less feature learning is not a large hindrance for least in the case of this combination of model, initialisation and dataset. This measure of features looks at hidden layer activations rather than the Jacobian of the network, so it also provides evidence that both interpretations (Neural Tangent Kernel vs. hidden layer representations) of *feature learning* used in the literature align.

In Figure 6, we plot the evolution of the first 3 filters for the first convolutional layer, for both $K=1$ and $K=5\times10^6$, with each column representing 10000 epochs. Each image shows the change in the filter up to that epoch and maps from [-0.5,0.5] to [0,1] in order to be plotted. We can clearly see that the filters evolve faster for $K=1$ and after training is finished for $K=5\times10^6$, the features are comparable to those of $K=1$ at about 20% of the way through training. This 20% matches when the linear probe for $K=1$ gets a similar test accuracy to the linear probe for $K=5\cdot10^6$ at the end of training.

This shows from two different measures that the intuitive idea that less frequent feature learning results in less feature learning too holds up in this case.

Finally Figure 7 connects the reduced feature learning with another intuitive definition of features. Here the setup is a 2-dimensional XOR problem with inputs $\mathbf{X}_i\in\mathbb{R}^2$ with $\mathbf{X}_i=\mathbf{z}_i+\epsilon_i$, $\mathbf{z}_{i1}$, $\mathbf{z}_{i2}\sim$ Rademacher, $\epsilon_i\sim\mathcal{N}(0,0.1\cdot\mathbf{I})$ and labels $\mathbf{y}_i=\mathbf{X}_{i1}\times\mathbf{X}_{i2}$. We train a single hidden layer network with different values of $K$ to convergence and plot the features and feature alignment.

The top plot is the *feature locations*, we define this as the rows of the first layer weights ($\mathbf{w}_{1i}$ for all $i$) which we plot in 2D space and colour by the corresponding second layer weight $\mathbf{w}_{2i}$

The second plot is a histogram of the *feature alignment* for all features (rows in the first layer weights). The feature alignment $a(\cdot)$ is the maximum cosine similarity with a cluster centre from the data.

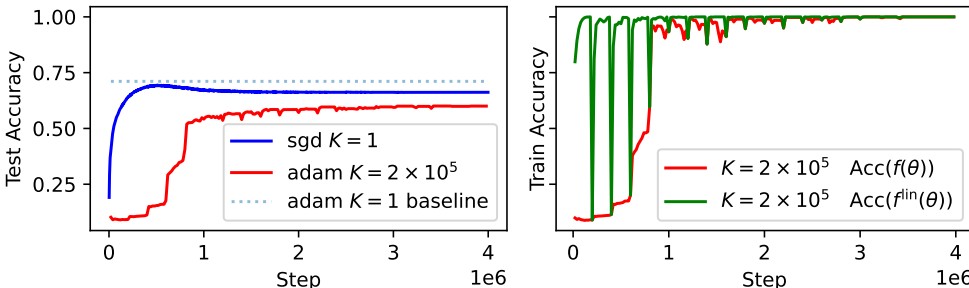

Figure 8: Iterative linearisation using Adam to completely optimise the convex problem each step. Here we compare standard gradient descent ($K = 1$) and a variant of iterative linearisation where the resulting convex objective from each linearisation is fully optimised ($K = 2 \times 10^5$) using Adam before re-linearising. In contrast to previous results, this results in worse performance ( 69% vs  60%). Adam by itself achieves over 70% on this task so the reduction in performance is due to fully optimising the linearised model and not due to swapping the optimiser.

The bottom two plots are similar but for varying $\lambda$ and we will discuss the similarities between the two further in the next section.

$$a(\mathbf{w}) = \max_{\mathbf{c} \in \{-1,1\}^2} \frac{\mathbf{w} \cdot \mathbf{c}}{\|\mathbf{w}\|_2 \cdot \|\mathbf{c}\|_2} \tag{11}$$

Each feature's location shows the input-to-hidden-layer weights and the colour represents the output weight; we do not plot the biases. Feature alignments are found per feature from the maximum cosine similarity between a feature location and any $[x, y]$, $x, y \in \{-1, 1\}$ and plotted on a histogram below. We can see the same degradation in feature quality with increased $K$ that we have seen previously, but now the actual features are easier to see explicitly. We also note that for such a simple case, the performance does not degrade with higher $K$ despite the poor quality features, as there are still enough well-located features to clearly distinguish classes with a simple linear classifier on the learned features.

These findings confirm that increasing $K$ reduces the *amount* of feature learning, not just the frequency, supporting the use of iterative linearisation as a tool for studying the impact of feature learning on generalisation.

### 4.3 Connecting to second-order optimisation methods

As covered in Section 3.3, iterative linearisation is closely related to second-order optimisation. To further validate the connection between damping and feature learning, we now perform iterative linearisation while numerically solving the convex softmax cross-entropy loss of the linearised neural network using Adam Figure 8. To make the graphs more readable we use a large constant $K$ rather than a varying $K$, intended to make sure that it reaches 100% train accuracy with each re-linearisation (see graph on the right) and that these are spaced regularly in the graph. We use Adam to solve the linear problem faster than would be possible using vanilla SGD for computational efficiency. Here the network converges to a solution which generalises much less well with 60.3% accuracy vs 69.4% for $K = 1$. In this case, the network has achieved a low enough train loss before it has a chance to learn features. This results in less feature learning and less generalisation.

We ensure that this is not due to using Adam by comparing to an Adam $K = 1$ baseline (dotted line), which achieves over 70%. $K = 2 \times 10^5$ also converges to the same test accuracy as using $K = 10^8$ with SGD in Figure 3. This gives confidence that we can safely swap SGD for Adam for the next experiment which would be completely infeasible to run with SGD.

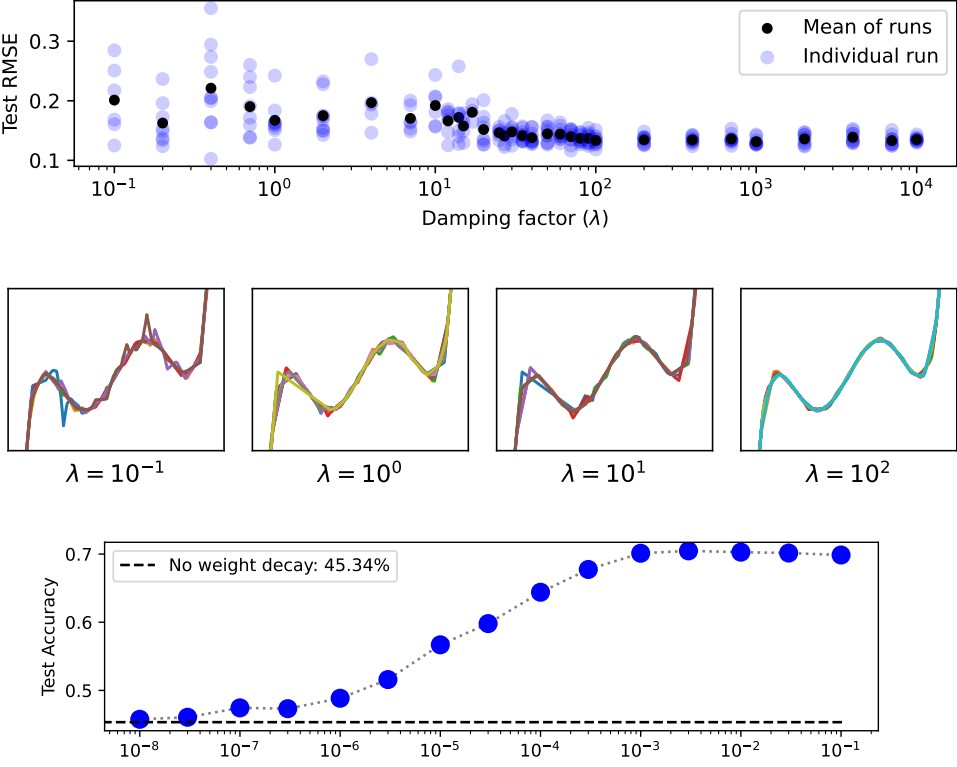

Figure 9: Improvement of generalisation as damping increases. The top plot shows test RMSE as damping is changed (runs with train RMSE over 2.5 are removed), with a reduction in the mean test RMSE and an increase in consistency — in particular note the number of high RMSE points with low damping factor despite the minimums being similar. The centre plot shows samples of the learnt functions as the damping parameter ($\lambda$) changes, where there is a clear trend towards smoother functions as damping increases. The bottom plot shows this more clearly, on CIFAR10 with a basic CNN. Here damping is simulated by modifying the weight decay on $\|\theta - \theta_0\|^2$ in iterative linearisation loss using Adam to fully solve each time as the full problem is both too large to add $\lambda I$ before inverting and no longer has a closed form solution due the the cross-entropy loss.

In Section 3.3, we also conjectured that there is a parallel between damping in second order methods and feature update frequency in iterative linearisation, with both impacting how far the weights can move to optimise the proxy model. To further investigate this connection, we show in Figure 9 that the increase in damping also improves generalisation. The top part of the figure examines varying damping on a 1D toy problem across many random initialisations. We can see that increasing damping improves test RMSE in worst case and average case performance, while best case remains fairly constant. Plotting example functions learnt for 4 different damping values in the middle graph shows visually how the learnt functions become smoother as the damping increases. Finally the bottom graph simulates damping when training a CNN with softmax, when we cannot simply invert the matrix. We do this by modifying the loss function to add a $\lambda \|\theta_s - \theta\|^2$ regularisation term (equivalent to damping with value $\lambda$ as shown in Section 3.3) and solving the linear least squares using Adam. Here the exact same phenomena is found where increased damping improves generalisation performance. We encourage the reader to compare the generalisation trend with reducing damping in the bottom plot of Figure 9 to the trend for increasing $K$ in Figure 3 which shows the beginning of a very similar trend before it becomes computationally infeasible for large $K$.

Finally, we refer back to Figure 7 (described in the previous section) where we compare the features learnt with varied values of $K$ and $\lambda$. We can clearly see that small $K$ and large $\lambda$ do a lot of feature learning and align well to the data. This alignment decreases as $K$ increases or $\lambda$ decreases. The exact ways that they

change are of course different but the general idea of less feature learning is clear. This provides further evidence for the parallel between these two methods.

These results provide strong empirical evidence for two important points. The first is the parallel between the feature learning impact of damping vs iterative linearisation periodicity with smaller values of $K$ being similar to more damping and resulting in increased feature learning. The second is that more exact second order methods may force us to add more damping to ensure good generalisation.

## 5 Related Work

The algorithm we introduce, *iterative linearisation*, is very similar to Truncated Gauss-Newton (TGN) as studied in Gratton et al. (2007) but with a very different focus which affects how it is used and some of the details. The algorithm differs in the condition of when to re-linearise — TGN uses a term to estimate if it is close to the solution and this is used to prove that under certain conditions it is close enough to Gauss-Newton to achieve local convergence. In contrast, iterative linearisation re-linearises every $K$ steps and explicitly tries to give a different solution to that of Gauss-Newton in order to adjust feature learning. Gratton et al. (2007) examine TGN from a mathematical perspective and show local convergence and convergence rates. We empirically examine iterative linearisation in deep learning settings and how it affects feature learning.

A number of papers have looked into feature learning with regards to the NTK regime. Fort et al. (2020) compare allowing feature learning initially before fixing the empirical NTK, showing that only a few feature learning epochs are needed to achieve similar performance. We build on this, reducing down to 10-15 feature learning updates to achieve comparable performance and show examples where this breaks down. Lewkowycz et al. (2020) obtains similar conclusions about feature learning happening early in training. In contrast, Vyas et al. (2022) look at the continuing changes of the NTK later in training, finding that there is some continued benefit in more feature learning under certain conditions. Our work builds on both of these, finding that there can still be room for improvements with more feature learning, but the improvements are very marginal after only very few feature updates.

Buffelli et al. (2024) proposes an exact way of computing the GN step efficiently in reversible neural networks and shows that this generalises less well. They consider small step sizes and find similar feature learning between GN and SGD with both less than Adam. In contrast, we consider the full GN step and find that this performs less feature learning than SGD until enough damping is added.

Allowing feature learning in the infinite width has received significant attention. Yang & Hu (2021) use a different parameterisation scaling to allow feature learning which performs better than finite networks, though it can only be computed exactly in very restricted settings. Feature learning has also been studied in the mean field limit (Mei et al., 2018; Chizat & Bach, 2018) on single hidden layer neural networks where, unlike the NTK limit, feature learning still occurs. There are a number of recent works looking at feature learning in this limited setting in a more theoretical way (Abbe et al., 2023; Bietti et al., 2023), including under a single feature update with isotropic (Ba et al., 2022) and structured data (Demir & Dogan, 2025; Hosseini et al., 2023). One relevant result from this line of research is that even a single gradient step can be enough to move to a feature learning regime. Our empirical findings align with this principle. While our experiments on deep networks do not find that a single update is entirely sufficient, we similarly observe that only a small number of feature learning updates are necessary to achieve the majority of performance gains. The quantitative difference is likely because our empirical setting, involving deep architectures and real-world datasets, is significantly removed from the idealized theoretical frameworks used in these papers.

Chizat et al. (2019) consider a different interpolation between finite and infinite networks, finding that as they get closer to their infinite width analogues, they perform less well empirically as they approach this limit. This interpolation is cleaner theoretically but more difficult to connect to feature learning, so we do not use it in this investigation.

Much of the second-order optimisation literature considers damping in detail. Typically, in order to improve the numerical stability of the matrix inversion (Dauphin et al., 2014; Martens, 2010) or optimisation speed (Martens & Grosse, 2015) due to trust regions of the proxy model, not understanding its effect on generalisation.

A number of results look at measuring feature learning empirically, but there is no accepted quantifiable measure of feature learning. Zeiler & Fergus (2014) is representative of a number of papers that use a similar approach of finding inputs that activate filters. While illuminating in some cases, this can be difficult to quantify precisely. A more quantifiable method is Alain & Bengio (2017) which uses *linear probes* to evaluate the hidden representation at different levels. Unfortunately, this has less direct connection to feature learning, only to the effectiveness of the current features, though we would expect these to be correlated. We use ideas from both of these directions to help understand feature learning in our models.

## 6   Conclusion

This paper has formalised *iterative linearisation*, a novel training algorithm that interpolates between gradient descent on the standard and linearised neural network as a parallel to infinite width vs finite networks. We justify it as a valid learning algorithm with reference to an intrinsic connection to the Gauss-Newton method. We show that by decreasing the frequency of feature updates within iterative linearisation, we can control the amount of feature learning during training, providing a powerful tool to help understand optimisation in neural networks.

In the case of datasets like CIFAR10, we show that a very small amount of feature learning is sufficient to achieve comparable test accuracy to SGD across a variety of settings such as full/mini-batch, use of data augmentations, model architecture and dataset size. We also show that *some* feature learning is required for good generalisation, connecting this with the fact that a fixed empirical Neural Tangent Kernel does not learn features and thus does not generalise well. This provides the important insight that while feature learning is necessary, SGD performance can be achieved with significantly less feature learning than expected. This supports the conclusion of not needing too many feature learning steps from Fort et al. (2020) in a different setting with an order of magnitude fewer feature learning steps but spread throughout training.

We connect the feature update frequency in iterative linearisation to damping in Gauss-Newton, providing a feature learning-based explanation for damping, backing this up with both theoretical insights connecting them and empirical observations showing the generalisation benefit of increasing damping or decreasing $K$. Our evidence for this insight includes almost identical theoretical behaviour at the extremes (Section 3.3) and similar behaviour empirically in the middle (Section 4.3), as well as an intuition connecting damping and periodicity $K$.

We note one major takeaway for practitioners from this work, which is to be aware of the feature learning capabilities of their algorithm and ensure that early training has enough feature learning. This includes larger damping values early in training if using second order optimisation, or saving on computation by doing less recalculation of features later in training if using an algorithm which can support that.

### 6.1   Limitations and Future Work

Due to computational complexity with the need for small learning rates, most experiments are a single run and on smaller architectures. As such, it is possible that these results do not generalise fully to transformers and larger models. For the same reason, we only use CIFAR10 and some low-dimensional datasets. Extending to transformers and more complex datasets would improve the rigour of this line of investigation.

There is no accepted way to measure feature learning quantitatively, so we make do with simplistic proxies. Better measurement tools or more in-depth analysis would help provide more evidence of our claim that increasing $K$ decreases feature learning.

In this paper, we only consider *fixed period* iterative linearisation, where we update the feature vector $\phi$ at regular intervals. However, Fort et al. (2020) showed that the empirical NTK changes faster earlier in training, so it makes sense for $K$ to be more adaptive if this approach is to be used to inspire more efficient training algorithms. In particular, when fine-tuning large models such as LLMs, there may be a way to improve efficiency by not always updating features in this way.

## Acknowledgments

*Temporarily removed for blind review*

## Broader impact statement

This is a paper looking at the underpinnings of generalisation in deep learning, as such it takes a small step towards improving the reliability and robustness of deep learning techniques. Additionally, insight into understanding feature learning could prove important in interpretability research and understanding model behaviour. This line of research also has the potential to result in improved training algorithms with all of the potential positive and negative societal consequences such as reduced energy consumption for large scale training and easier access to advanced models for both helpful and nefarious purposes. We acknowledge that these risks and benefits are shared across foundational deep learning research..

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

# A  Iterative linearisation with a general loss function

In Section 3 we show how to get to iterative linearisation from standard gradient under mean squared error loss. The use of mean squared error is more instructive due to its similarities with NTK results, however it is not strictly necessary. For completeness we include here the same idea but for a general loss function $\mathcal{L}(\cdot)$.

Standard gradient descent on a function $f_\theta(\cdot)$ parameterised by $\theta$, with step size $\eta$ and data $X$ can be written as

$$\theta_{t+1} = \theta_t - \eta \nabla_\theta \mathcal{L}(f_{\theta_t}(X))$$

We can apply the chain rule, resulting in

$$\theta_{t+1} = \theta_t - \eta \nabla_\theta f_{\theta_t}(X) \mathcal{L}'(f_{\theta_t}(X))$$

Where $r(\cdot)$ is the derivative of $\mathcal{L}(\cdot)$ (in the case of mean squared error, this is the residual: $r(\hat{Y}) = \hat{Y} - Y$). Now again using $\phi_t = \nabla_\theta f_{\theta_t}(X)$, we can write this as

$$\theta_{t+1} = \theta_t - \eta \phi_t r(f_{\theta_t}(X))$$

With a similar argument to Section 3, we note that we don't need to update the features $\phi_t$ every step, resulting in the following formulation.

$$\theta_{t+1} = \theta_t - \eta \phi_s^{\text{lin}} r\left(f_{s,t}^{\text{lin}}(X) - Y\right) \tag{12}$$

$$\phi_s^{\text{lin}} = \nabla_\theta f_{\theta_s}(X) \tag{13}$$

where $s = K * \lfloor \frac{t}{K} \rfloor$

This now lets us use softmax followed by cross-entropy in the loss $\mathcal{L}(\cdot)$ while maintaining the same interpretation, as we do for the CIFAR10 results.

# B  Proof of closed form solution of gradient flow

**Lemma B.0.0.** *Gradient flow on a linear model $f_\theta(x) = \phi(x)^\top \theta$ with squared error loss $\mathcal{L}(Y) = \frac{1}{2}\|Y - \hat{Y}\|_2^2$ starting at weights $\theta_0$ converges to $\theta_0 - (\phi^\top \phi)^{-1} \phi^\top r(f_{\theta_0}(X))$, when $\phi^\top \phi$ is invertible. Where $\phi = \nabla_\theta f_{\theta_0}(X)$ is the Jacobian on the dataset $X$ and $r(Y) = (Y - \hat{Y})$ is the loss residual.*

*Proof.* The differential equations defining how $\theta_t$ and $f_{\theta_t}$ change are given by.

$$\dot{\theta}_t = -\phi^\top r(f_{\theta_t}(X)) \tag{14}$$

$$\dot{f_{\theta_t}}(X) = -(\phi\phi^\top) r(f_{\theta_t}(X)) \tag{15}$$

The evolution of the function only depends on the residual so the differential equation can be solved in closed form as follows by substituting $z_t = r(f_{\theta_t}(X))$.

$$\dot{z}_t = -(\phi\phi^\top) z_t \tag{16}$$

$$z_t = e^{-(\phi\phi^\top)t} z_0 \tag{17}$$

$$f_{\theta_t}(X) - Y = e^{-(\phi\phi^\top)t} r(f_{\theta_0}(X)) \tag{18}$$

$$f_{\theta_t}(X) = Y + e^{-(\phi\phi^\top)t} r(f_{\theta_0}(X)) \tag{19}$$

Solving for the weights can be done using the solution to $f_{\theta_t}(X)$ above.

$$\dot{\theta}_t = -\phi^\top r(f_{\theta_t}(X)) \tag{20}$$

$$= -\phi^\top e^{-(\phi\phi^\top)t} r(f_{\theta_0}(X)) \tag{21}$$

$$\theta_t = \phi^\top \left(\phi\phi^\top\right)^{-1} e^{-(\phi\phi^\top)t} r(f_{\theta_0}(X)) + C \tag{22}$$

$$\theta_t - \theta_0 = -\phi^\top \left(\phi\phi^\top\right)^{-1} \left(I - e^{-(\phi\phi^\top)t}\right) r\left(f_{\theta_0}(X)\right) \tag{23}$$

$$\theta_\infty = \theta_0 - \phi^\top \left(\phi\phi^\top\right)^{-1} r\left(f_{\theta_0}(X)\right) \tag{24}$$

This solves it in a kernelised regime. If we assume that $\left(\phi^\top\phi\right)$ is invertible then we can write $\phi^\top \left(\phi\phi^\top\right)^{-1} = \left(\phi^\top\phi\right)^{-1} \left(\phi^\top\phi\right) \phi^\top \left(\phi\phi^\top\right)^{-1} = (\phi^\top\phi)^{-1}\phi^\top$ to get the standard formulation of $\theta_0 - (\phi^\top\phi)^{-1}\phi^\top r\left(f_{\theta_0}(X)\right)$

$\square$

## C  Further experiments and experimental details

### C.1  Further Experiments

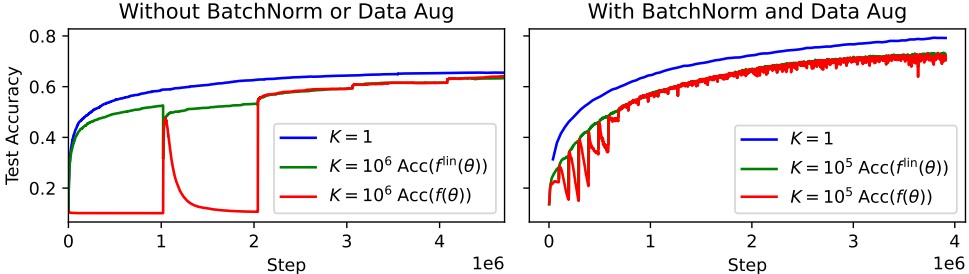

Figure 10: ResNet18 runs with and without BatchNorm and data augmentation. Large $K$ iterative linearisation again achieves similar test performance to SGD. Runs with BatchNorm are far more likely to diverge due to linearising the BatchNorm layer hence why $K$ is much smaller for that run.

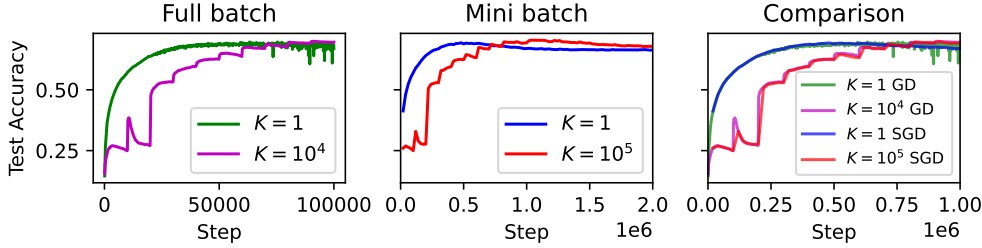

Figure 11: Full batch and mini-batch iterative linearisation for various values of $K$ on a standard CNN architecture on CIFAR10. The left and centre plots compare iterative linearisation to standard training on full-batch and mini-batch gradient descent, respectively. The full batch runs use a learning rate of 1e-3, whereas the mini-batch is scaled down to 1e-4 for stability. As such, we scale $K$ up by a factor of 10, too. The steps of the full batch runs are similarly scaled down by a factor of 10 for the comparison plot on the right.

We show in Figure 10 the results for a ResNet18. While these clearly work similarly, they are computationally expensive, and the one with BatchNorm would require a lot of tuning of $K$ and $\eta$ and many days of training

to get as clean graphs as we have for a simple CNN. We include them for completeness and as evidence this behaviour generalises.

We also include in Figure 11 a comparison between full batch and mini batch training for CIFAR10 with our simple CNN setup. This covers a gap between our full batch theoretical work and the practical mini batch results.

## C.2 Experimental Setup

### C.2.1 Simple CNN experiments

All CIFAR10 experiments (except those with ResNets which used a ResNet-18) use a modified variant of LeNet where both convolutional layers have 50 channels. This results in two convolutional layers, each with kernel size 5 and 50 channels and max pooling after each. These are followed by dense layers of sizes 120 and 84 and an output layer of size 10. All inner activations are ReLU and the output layer uses softmax. This was chosen as simply a slightly larger and more modern version of LeNet. Learning rates are given in plot captions and batch sizes were either 256 or full batch. For Figures 3 and 4, optimal early stopping was used.

### C.2.2 ResNet18 experiments

For the ResNet results in Figure 10, we show performance on a ResNet-18 without BatchNorm or data augmentation (left) with a learning rate of 7e-7 and $K = 10^6$ and on a ResNet-18 with BatchNorm and data augmentation (right) with a learning rate of 1e-5 and $K = 10^5$.

### C.2.3 MLP experiments

All 1 dimensional regression tasks are trained on a quintic function $(\frac{1}{32}x^5 - \frac{1}{2}x^3 + 2x - 1)$ — chosen for having non-trivial features and bounded outputs in the [-4, 4] range — with 20 uniformly spaced datapoints used for training data and 1000 uniformly spaced datapoints for the test data. The neural network was a 5 layer MLP with 50 neurons per layer with ReLU activations and squared error loss. They were trained full batch through the Gauss-Newton algorithm described in Algorithm 1 using the given $\lambda$ values for damping.

