# OpenReview forum: "Understanding Sparse Feature Updates in Deep Networks using Iterative Linearisation"
_TMLR — Rejected by TMLR_

### Review · Reviewer_e1Lp · 2025-05-16

**Summary Of Contributions:**

This paper investigates the relationship between feature learning and test performance in neural networks. To this end, it introduces a novel algorithm, Iterative Linearisation (IL), parameterized by a hyperparameter $K\in\mathbb{N}$, which denotes the number of weight updates between two gradient computations. IL interpolates between Neural Tangent Kernel (NTK) training ( $K=\infty$, where gradients remain fixed at initialization) and standard stochastic gradient descent (SGD) ($K=1$, where gradients are updated at every step).

The paper further highlights a close connection between IL and the Gauss-Newton (GN) algorithm: when $K$ is large but finite and the step size is small, IL approximates a GN update on the linearized model. This builds on the well-known result that gradient flow in linear models with mean squared error (MSE) loss converges to the GN solution. Accordingly, IL can be interpreted as performing a GN-like update between every $K$ steps.

Additionally, the paper examines how damping (L2 regularization) interacts with feature learning, with the extent of learning controlled by $K$.

A series of small-scale experiments are presented to evaluate the performance of IL and to draw insights into its behavior.

**Audience:**

Yes

**Broader Impact Concerns:**

This is a paper contributing to deep learning theory and I see no concerns on the ethical implications of the work that would require adding a Broader Impact Statement.

**Claims And Evidence:**

Yes

**Requested Changes:**

I think the paper needs to address to the followings:

1. Reposition this paper. it is quite obvious that more gradient updates means more feature learning, and that depending on datasets, one might not need too many gradient updates. Then what should be the goal of this paper? Does it want to discover more non-trivial phenomena related to feature learning? Or does it want to explore the efficacy of IL algorithm in practical settings? For instance, how much  computing time can be saved when we update the gradients less often? What is the trade-off between computing time and test performance? Does IL also work with optimizers other than SGD and Adam? Alternatively, does the paper want to investigate the theoretical sides of IL? Does IL lead to other implicit bias than vanilla SGD or kernel learning?
2. Experiment with sweeps on both damping $\lambda$ and gradient update frequency $K$, and present the train and test accuracy in grid plots.

**Strengths And Weaknesses:**

Iterative Linearisation (IL) is a nuanced idea to connect NTK training and conventional SGD training. The relationship of IL to GN algorithm is also insightful.

However, this paper falls short in the followings perspective:
1. The experiment seems insufficient for the claims. For example, the claim that "the parallel between the feature learning impact of damping vs iterative linearisation periodicity with smaller values of K being similar to more damping and resulting in increased feature learning" is barely supported by any plots or figures. Figure 9 shows increasing damping improves generalization, which is already known in both NTK and SGD training, and has nothing to do with IL periodicity $K$. The only plots showing the effect of varying $K$ are Figure 3 on test accuracy and Figure 7 on feature alignment, which are both irrelevant to the context of damping.
2. I gently disagree with some of the experimental setup. For instance, in Figure 9, the paper switches from SGD to Adam on a convex softmax cross-entropy loss of the linearised neural network. Although Adam can accelerates the training, it might be inconclusive on the claims about damping: since Adam uses coupled weight decay which might have non-trivial interactions with the test accuracy. Using AdamW could be a better choice.

Hence, the contributions of this paper are rather 1) intuitive or known, like more gradient updates means more feature learning; 2) or incomplete, like the relationship between gradient updates frequency $K$ and equivalent damping $\lambda$.

---

> ### Author Response · Authors · 2025-07-16
>
> Thank you for this review and pushing us to be more thorough with this work. We apologise for the delay in responding.
>
> Your primary concern was that the parallel between damping (λ) and IL periodicity (K) was insufficiently supported. To address this directly, we have performed a new experiment, now included in an updated Figure 7, which creates a side-by-side visual comparison of the features learned when sweeping over K and λ. We believe this new evidence provides the direct support you requested and makes the connection explicit. We address your specific points below
>
> Weakness 1:
> * We are not aware of published results stating that damping is important for generalisation. If you have relevant references we would greatly appreciate these to include in our discussion. While damping is used in second-order methods for numerical stability, its role as a direct mechanism to improve generalisation (analogous to regularisation) is not well-established, and we believe our analysis of damping's role in generalisation is a novel contribution in its own right, before even drawing the parallel to IL. There has been some work on Gauss-Newton (Bufelli et al, 2024 in Neurips) which performed damping through a learning rate and some KFAC results (Ma et al, 2020 in AAAI) showing that good values of damping and learning rate are positively correlated but not looking at generalisation specifically. None of this has shown the simple direct impact of damping on generalisation in neural networks.
> * However, our main contribution in the referenced section and why we mention damping, is to draw the parallel between this effect and the update periodicity K. This provides a new, feature-learning-based explanation for why damping helps generalisation, moving beyond its traditional role of ensuring numerical stability. We have strengthened this claim in the paper by explicitly stating that while we don't offer a formal proof of equivalence, the very similar behavior at the extremes (K=1 (gradient descent) vs large λ (gradient flow); large K (truncated Gauss-Newton vs λ=0 (Gauss-Newton)) and the similar empirical behavior in between (Figure 3 vs Figure 9) provide strong evidence for this parallel. We also note that there may be a misunderstanding here as damping in SGD is not discussed, we show an equivalence between damping and regularisation but note that that is a regularisation term of the distance from the previous GN step/relinearisation point, not a standard regularisation term (of which there are of course many results regarding generalisation).
> * We agree that we can make the connection between K and lambda more clear however. While the best comparison is out of reach for computational reasons, we have replicated the setup of figure 7 but varying the damping in Gauss-Newton as a comparison. Additionally, we have added a summary of our main arguments for this point in our conclusion.
> Weakness 2:
> * We are training a strictly convex model until convergence so the only difference here will be the time until convergence with Adam vs AdamW (assuming settings such that they both converge). If we had no damping (the dotted line in figure 9) then it is merely convex and there may be multiple solutions with 0 loss and hence Adam may converge to a different one. In the rest of our results, the landscape is strictly convex (a convex function plus the strictly convex regularisation is strictly convex) and they should converge to the same solution.
> * However your point is well taken and AdamW would be a nicer option here, unfortunately AdamW doesn’t work for this out of the box as it regularises the parameters instead of the distance of those parameters from a custom (and non-constant) point. We have now rederived and reimplemented AdamW for this setting however. After which, we reran for a damping value of 0.01 and got the test accuracy within 0.2% (apologies for not redoing the entire graph, it takes a while to generate and our mathematical argument already says that this should make no difference, we will regenerate the full graph in the background if you still believe it is helpful and update that before the camera-ready).
>
> Continued in next comment

---

> > ### Author Response · Authors · 2025-07-16
> >
> > Addressing your points about the utility and convincingness of our contributions:
> > * We appreciate the point that the connection between gradient updates and feature learning is intuitive. However, as a TMLR submission, a key goal of our work is to move beyond this general intuition and provide a more quantitative, and at times surprising, picture. For instance, a core challenge in this area is that 'feature learning' lacks a single, agreed-upon definition. In Section 4.3, we address this by using multiple distinct proxies (linear separability, filter visualisation) to measure it. Our results show a non-trivial relationship: IL with a moderate K can achieve comparable generalisation to SGD with orders of magnitude fewer gradient updates, suggesting that not all updates contribute equally to learning useful features. This nuanced finding, we believe, is a step beyond the initial intuition.
> > * With regards to the "incompleteness" of the relationship between gradient updates frequency K and equivalent damping lambda. We agree that we have not completely proven this claim (nor do we claim to have, we only claim to have shown a parallel between them). We have instead shown that they are equivalent at the extremes and provided empirical evidence that they both appear to have the same impact on training. While it would be worth proving this completely, our current method does not scale enough to do so empirically and we believe that what we have is already of interest to the community.
> > * Our first (and main) listed contribution of formalising iterative linearisation as a method to investigate feature learning you already state as a strength in your response.
> >
> > With regards to the requested changes:
> >
> > Point 1:
> > * Thank you for pushing us to clarify the paper's core objective. To directly answer the question: the primary goal of this paper is to introduce and leverage Iterative Linearisation (IL) as a precise methodological tool for investigating the impact of feature learning. Our aim is not to propose IL as a new state-of-the-art training algorithm, but rather to use its periodicity parameter, K, as a 'knob' to carefully control the amount of feature learning between the two extremes of a feature-learning method (like SGD) and methods such as NTK which don’t learn features or Gauss-Newton which learn less features. By tuning this knob, we can explore non-trivial phenomena, such as the correlation between feature learning and test performance, and better understand the feature learning impacts of second order methods such as Gauss-Newton. We have revised the introduction and conclusion to state this methodological goal more explicitly.
> > * Our two secondary contributions are insights to demonstrate the effectiveness of this technique. It gives a clear way to control feature learning giving insights about the amount of feature learning needed for good performance and additionally connects feature learning/generalisation to damping in second order optimisation.
> >
> > Point 2:
> >
> > Thank you for this suggestion to make the comparison clearer. We want to ensure we address it correctly and consider two possible interpretations of the 'grid plot' request.
> > * One interpretation is a single plot from a sweep over pairs of (λ, K). This is not straightforward, as λ and K are parameters for different conceptual methods (damped GN and IL, respectively) and combining them is not well-defined. While such a combination could be made, it would be very computationally expensive to perform a wide enough sweep and the interpretation of any results would be difficult as these methods are not necessarily additive in impact.
> > * Therefore, we believe a more likely and insightful interpretation is a figure that arranges separate sweeps over K and λ into a comparative grid of line plots, to make their parallel behavior visually explicit. We agree this is an excellent way to support our claim. We already have the comparison of Figures 3 and 9 for larger scale training but unfortunately large K are intractable and we cannot have the full sweep here. As mentioned in our response to your first point, we have now updated Figure 7 to include a sweep over λ alongside the original sweep over K, creating exactly this kind of direct visual comparison. We believe this new figure provides the clear, comparative evidence you requested and strengthens our claim of a parallel between the two methods.

---

### Review · Reviewer_Eusn · 2025-06-08

**Summary Of Contributions:**

The authors consider *iterative linearization* as an approach that lies between linear NTK and non-linear SGD.
This method consists of periods of linear updates as in the NTK setting where the parameter $K$ governs the length of each period, so that $K=1$ corresponds to SGD and $K=\infty$ to NTK. The authors then study the behavior for finite $K$ with a focus on larger $K$, in which case the algorithm can be connected to the Gauss-Newton Algorithm (for minimizing an l2 objective).
They experimentally analyze their proposed method on some simple datasets and networks.

**Audience:**

Yes

**Claims And Evidence:**

No

**Requested Changes:**

Major Concern:
The main concern is a lack of clarity, especially when a clear (mostly experimental) analysis of the proposed algorithm is the main contribution. Here are some problematic examples of this:
- In section 3.2 the third paragraph claims to just have proven the connection to Gauss-Newton while the previous paragraph is a heuristic that is not formalized, but rather stated in words.
Related to this in Figure 1, which comes before, it is claimed that "For very large K, it will train until convergence for each linearisation
giving the same results for all larger K — this is equivalent to the Gauss-Newton algorithm". I do not believe this statement to be correct in this form as for all larger $K$ would mean no difference between different $K$, instead I would assume this to be similar. Also there should be a reference to where this is proven as it is not shown in the figure (so perhaps putting it in the text referring to Figure 1).
- In section 4 it is not clear what the "simple CNN" network is, making it hard to judge the experiments.
- Figure 3 is hard to read visually: the legend overlaps with the plots, the y axis is cut off at 0.2 so the x-axis before $10^3$ is not very useful, also how do K = $10^4$ and $10^5$ start at a non-trivial test loss not clear how one should read this.
Also in the description of Figure 3 it is claimed that "At $K=10^8$, this simulates the Gauss-Newton algorithm accurately, bringing the
train loss to almost 0 each linearisation." There is no train loss plotted in this Figure. Also how would one be able to verify this based on some plots, simulating an algorithm is not the same as achieving similar performance on derived metrics.
- A general comment about section 4, while of course it is not reasonable to expect all experiments to be done on SOTA models, for a mostly  experimental paper it seems a bit strange to have no experiments on some model that is close to optimal performance. One could even consider having some pretrained model, fix the initial layers and then train on top, while not the most realistic would be simple enough to implement. For example one can get higher performance on more complicated datasets by doing this, even in the case of adding MLP on a pretrained Resnet.

Minor Points:
- Figure 4 what network architecture is used? Also it would be interesting to see at which $K$ it starts declining towards NTK
- There is a lot of talk about large $K$ being close to GN, why is there no experiment directly comparing GN to the proposed algorithm.
- In Figure 7 part of the description is in the text referring to Figure 7, the explanation below the Figure should explain this. In particular the sentence "The feature location is the first layer weight rows ($w_{1i}$ for all i) and is coloured by the corresponding second layer
weight $w_{2i}$." should be made clearer by referring to the first row. As it is, it is not immediately clear what "feature location" is supposed to mean
- In appendix C last paragraph "They were trained full batch through the Gauss-Newton algorithm described using...", would be good to add a reference where the GN was described


Overall the Idea seems interesting, but the current write up is not clear enough, I would encourage the authors to change the writing in such a way that the above concerns and questions are resolved. In particular the following two main points:
- For every claim made, make it clear where it is proven/derived from and make the presentation more clear
- Update the experimental section to be more clear, such that in particular the concerns above are addressed

**Strengths And Weaknesses:**

Strengths:
- The general question of finding an interpolation between SGD and NTK is well motivated
- The approach allows for a simple analysis and connection to Gauss-Newton for large $K$.


Weaknesses:

There is a major problem with clarity in this paper see Requested Changes below, in short:
- Many claims that are made are not (sufficiently) supported
- There is no coherent "train of thought" at times which makes it hard to follow the authors, which exacerbates the above problem

---

> ### Author Response · Authors · 2025-07-16
>
> Thank you for the detailed review. We apologise for the delay in responding.
>
> Thank you for the points about clarity, especially the examples. We have done a thorough review to try to find any other examples of ambiguity and unsubstantiated claims, as well as fixing your suggestions as follows:
> * GN equivalence claim: You’re right, this was perhaps overly concise. Let us expand on it here (we have also updated the paper to make this clearer). The ‘exact’ there refers to an exact solution of the linearised model (this is GN by definition), clearly this was confusing so we have rewritten that section to make it more explicit what claims we are making. GN requires solving a linear equation, rather than doing this through a matrix inverse, this is often performed through simply running gradient descent until convergence (we have expanded our discussion of this point in section 3.2). Of course it may never converge perfectly, however the cited paper (Gratton et al, 2017) shows that “Truncated Gauss Newton” where this equation is only solved within a tolerance maintains the local convergence guarantees of GN. The proof of local convergence of iterative linearisation is later in this subsection. With regards to equivalence to GN, it becomes a question of what we mean by equivalent. This way of solving the least squares problem iteratively is the GN algorithm when used often in practice and according to most of the literature so we stuck with that lexicon. This appears to be confusing however so we have swapped to stating explicitly that this is Truncated Gauss-Newton that it is equivalent to to be more precise here.
> * “Simple CNN”: At the start of Section 4 we refer to the Appendix where this is described. However this clearly didn’t stand out enough, so we have linked it in parenthesis at the first mention and made a heading in the appendix where we describe it, as well as using the exact phrase “simple CNN” in the description so it is searchable. Additionally the first use at the start of the results section now has a very short description “a modified LeNet with extra channels”.
> * Figure 3: These are some very valid criticisms, we have improved it as follows:
>   * We have fixed the scale and legend overlap
>   * The accuracy is not meaningful when not a multiple of K for the case of this plot (we show the full trajectories of iterative linearisation runs in figure 2 but that would make this plot unreadable and misleading). As such the first point is at K steps (which is why it ‘starts’ at non-trivial accuracy). Adding the accuracy at step 0 is difficult on a log-scale, as an alternative we now show it dropping vertically to the accuracy with the initial parameters. This isn’t perfect so we have expanded the discussion to make clear what is happening here.
>   * Train loss: you are correct, this was too strong a claim based on our evidence and we are not convinced it was correct. It has been corrected and we have checked for other unsupported points.
> * SOTA models: We completely agree that results on larger models would be great. However, we believe that we have already found very interesting insights with our method that are interesting on their own.Our method unfortunately doesn’t scale to SOTA models as we mention at the start of Section 4. The suggestions about training part of a model are interesting and in many contexts would be a great alternative, unfortunately the earlier layers will have already done a lot of the important feature learning (all of it if we only train the last layer, almost all if we add an MLP to the end) and as such there wouldn’t be much difference as modifying K changes the amount of feature learning. If there is a good representation already then a bit more or less feature learning isn’t very noticeable.
>
> Minor points:
> * This is the same simple CNN we are using for most experiments, thank you for pointing out that we don’t mention it! That’s fixed now. We would also want to see where it starts approaching NTK, unfortunately these runs are incredibly slow and resource intensive. The run for K=10^8 in the figure above took several weeks. To help address this query though, we’ve mentioned that run in the caption as it starts to answer your question.
> * We think this might be a clarity issue again unfortunately. Running Adam until convergence on each linearisation with no weight decay is GN (or at least the most common way of approximately scaling it - technically Truncated GN according to Gratton’s terminology ). However this is good to call out - this should absolutely be on this graph too, not just figure 9. It is now added to figure 3.
> * Thank you, we agree that the mathematical description in the caption is more precise than the textual one and have copied the same format there.
> * Fixed
>
> In summary, we thank the reviewer for this exceptionally detailed and helpful feedback. We believe the resulting changes have substantially improved the clarity, precision, and readability of our paper.

---

### Review · Reviewer_6dvT · 2025-06-21

**Summary Of Contributions:**

The submission introduces iterative linearization, a novel training algorithm that interpolates between full stochastic gradient descent (SGD) and lazy training (e.g., Neural Tangent Kernel regime) by periodically re-linearizing the network’s forward function. This method enables explicit control over feature learning frequency—a key determinant of generalization performance in deep networks.
The main contributions are as follows:
1. Iterative linearization decouples feature learning from learning rate and width by freezing features for a tunable number of steps $K$, then re-linearizing. This provides a controlled experimental setting to study the inductive benefits of feature updates.
2. The authors empirically show that as few as 10–15 feature updates during training can yield generalization performance comparable to full-feature-learning SGD on standard image benchmarks. This result refines prior findings (e.g., Fort et al., 2020) and suggests that the bulk of generalization benefit from feature learning can be achieved with minimal updates.
3. Through experiments varying dataset size, the authors demonstrate that all finite $K$ models (i.e., any with feature learning) scale better with data than purely lazy (NTK-like) models, consistent with theoretical results on the sample complexity gap between kernel and feature-learning models.
4. The paper draws a novel connection between the feature update frequency in iterative linearization and damping in Gauss-Newton optimization, showing that reduced damping (or smaller $K$) promotes more feature learning and better generalization. This reframes damping as not only a stability tool but also a feature learning regulator.
5. While this work focuses on training-induced feature learning, its results can be complementary to recent theoretical studies showing that structured input data can itself induce feature learning under small learning rates. This positions iterative linearization as a flexible method to probe both optimization- and data-driven routes to escaping the lazy regime.

**Audience:**

Yes

**Broader Impact Concerns:**

No critical ethical concerns arise from this work. However, the authors should consider expanding the Broader Impact Statement to discuss:
* The potential positive implications for more efficient model training and tuning, especially in large-scale settings,
* The dual-use risks associated with making training more efficient or generalizable,
* The role of this work in improving theoretical transparency and understanding of model behavior.

**Claims And Evidence:**

Yes

**Requested Changes:**

1. Acknowledge and position relative to data-driven feature learning work
    * The submission should explicitly discuss recent results showing that structured input data (e.g., low intrinsic dimension or Gaussian mixtures) can enable feature learning under small learning rates (e.g., Ma et al., 2022; Misiakiewicz et al., 2023, Mousavi-Hosseini et al 2023, Demir and Dogan 2025).
    * This would clarify that feature learning is not only training-driven, and would prevent misinterpretation that this work contradicts those findings.
2. Clarify the scope and limitations of the proposed framework
    * More discussion is needed on where iterative linearization may or may not apply, particularly regarding:
        * Architectures with normalization layers (e.g., BatchNorm, LayerNorm),
        * Tasks beyond image classification (e.g., NLP, transformers), and
        * Limitations due to small learning rates and convergence speed.
3. Provide theoretical insight or intuition on sample complexity scaling with $K$
    * While full proofs aren’t necessary, even a sketch or intuition for how sample complexity might interpolate between NTK and SGD as a function of $K$ would strengthen the contribution.
4. Introduce a discussion or rule-of-thumb for how practitioners might set K  K based on factors like dataset size, architecture depth, or training budget. Alternatively, mention potential for adaptive or early-phase tuning strategies (e.g., more frequent updates early in training).
5. Improve notational clarity:
    * Some notation around $\phi$,  $\theta$, and linearization steps could be streamlined or more explicitly defined, particularly for readers less familiar with Gauss-Newton.

**Strengths And Weaknesses:**

Strengths
1. Introduces iterative linearization, a new and insightful framework that interpolates between lazy training and full feature learning using a tunable periodicity parameter $K$. Also, offers a concrete mechanism to study and manipulate feature learning frequency, decoupled from learning rate or network width.
2. Empirically, it provides strong and consistent empirical evidence that only a small number of feature updates are sufficient for good generalization, and shows performance scaling with data size across a wide range of $K$ , validating theoretical predictions about sample complexity gaps.
3. Establishes a compelling theoretical and practical link between iterative linearization and (Truncated) Gauss-Newton optimization, offering new insights into how damping and precision relate to generalization via feature learning.
4. Experiments are thoughtful and varied (e.g., CNNs, ResNets, toy data, XOR), with use of linear probes, feature visualization, and kernel comparisons to diagnose feature learning dynamics.

Weaknesses
1. While the paper draws useful analogies and convergence arguments, it lacks formal guarantees or bounds on generalization or sample complexity as a function of $K$, which would elevate the impact. No formal characterization of the function classes learnable under different $K$ regimes.
2. Most experiments are on relatively standard CNNs and image classification tasks. It remains unclear whether the conclusions extend to transformers, LLMs, or other modalities, where feature learning may behave differently. BatchNorm and softmax layers are handled heuristically, and their interaction with re-linearization could be further clarified.
3. The paper focuses entirely on training dynamics as a driver of feature learning. Recent theoretical work shows that structured data alone can induce feature learning, even with small learning rates. This paper could benefit from explicitly acknowledging and situating itself relative to that line of work.
4. While the framework introduces $K$ as a useful knob, it offers no principled guidance for choosing it, especially for practitioners or real-world applications. Adaptive or data-driven strategies for setting $K$ are mentioned as future work but not explored.

---

> ### Author Response · Authors · 2025-07-16
>
> Thank you for your review and insights, and apologies for the delay in our response.
>
> Responses to requested changes (all weaknesses mentioned seem to have a matching requested change so we respond to those)
> 1. Acknowledge and position relative to data-driven feature learning work: Thank you for these valuable references. We have updated our related work section to situate our findings with respect to Mousavi-Hosseini et al. (2023) and Demir and Dogan (2025). We appreciate the pointer to Misiakiewicz et al. (2023); while we found it to be a broad survey, we believe you may have been referring to the highly relevant work from Abbe et al. (2023) (for which Misiakiewicz is an author), which we already discuss. We had difficulty locating 'Ma et al., 2022' based on the name and year alone; if the reviewer could provide a full reference, we would be grateful to include a discussion of it in our final version.
> 2. Clarify the scope and limitations of the proposed framework: This is an important point. We have expanded Section 6.1 ('Limitations and Future Work') to explicitly address the limitations as you suggested, though we are unable to make statements about how our method extends to settings which are too computationally expensive for us to test. We have also added a clarifying sentence to the introduction to better frame the scope of our analysis for the reader from the outset.
> 3. Provide theoretical insight or intuition on sample complexity scaling: This is an excellent question. While a formal derivation is beyond our current scope, we agree that providing intuition is valuable. We have added a discussion to the analysis of Figure 4 that explicitly articulates the intuition: feature learning provides a sample efficiency benefit, but only up to a point. At the extremes, very large K (approaching GN) requires more samples to achieve the same performance as moderately-sized K.
> 4. Introduce a discussion or rule-of-thumb for how practitioners might set K: This is a key question for translating this work into practice. While our primary goal is to use iterative linearisation as an analytical tool rather than a replacement for SGD, our findings do offer a general takeaway for practitioners: methods that encourage strong feature learning, such as SGD or second-order methods with increased damping, are likely most beneficial in the early phases of training and methods later in training can reduce feature learning to improve efficiency. We have expanded on this in our conclusion where we previously only mentioned variable periodicity schedules.
> 5. Improve notational clarity: We were following the notation used in the literature but agree that more consistent notation between Gauss-Newton and NTK is beneficial here, we have used $\phi_t = \nabla_\theta f_{\theta_t}(X)$ more consistently (except for in the introduction to keep that readable without reading any of the rest of the paper) and added a notation subsection with definitions of the most important notation we use. We have made other minor changes to improve clarity with this too.
>
> We have made changes to the Broader Impact Statement as requested.

---

### Decision · Action_Editor_GTw1 · 2025-07-28

**Recommendation:** Reject

**Additional Comments:**

The paper proposes an iterative linearization that smoothly interpolates between the linear NTK regime (lazy, no feature learning) and non-linear SGD (with feature learning). The idea is to re-linearize the predictor coming from the network after a tunable number of steps $K$, the extremes being $K=\infty$ which corresponds to the NTK regime and $K=1$ which corresponds to standard SGD. The authors then propose a connection between this iterative linearization and the Gauss-Newton algorithm.

All reviewers generally appreciated the idea, finding it insightful and original. However, they also pointed out a number of flaws, which were only partially addressed by the authors during the rebuttal process. One important concern raised by both Reviewer Eusn and Reviewer e1Lp is that the paper fails to position itself either as a theoretical or an empirical study. In fact, on the one hand, it lacks theoretical guarantees and, on the other hand, it lacks empirical evidence on larger models and the high-level ideas do not appear to be explored at a sufficient level of detail. Clarity also remained an issue even after the rebuttal. Some specific points raised by Reviewer Eusn are the following:

* For practical considerations the learning rate needs to be specified in addition to $K$. This is not always made clear enough.

* In Figure 3, the performance drops at $K=10^7$ but in Figure 4 below (which is on showing something slightly different) the performance at $K=10^7$ varies with dataset size and it actually becomes better in some cases. It is not clear what conclusion should be drawn here, does data size play a role?

* In Figure 5, the authors want to answer the following question "This raises the question of when iterative linearisation results in less overall feature learning versus simply clumping the feature learning from $K$ steps into $1$". But then $K = 1$ is only compared to $K=10^5$. Comparison across multiple values of $K$ is needed to reach a definitive conclusion about the trend.

* Figure 6 is only based on visual representations, but it is claimed that the filter is similar. This is not so obvious, as human visual perception may be an inaccurate estimator for other metrics like $L_2$ or other losses.

* Figure 7 now shows a new type of plot but on different data, so it is hard to compare it to the previous results.

In light of these issues, unfortunately I cannot recommend acceptance at this stage. I do think that the idea has value, so I would recommend the authors to re-submit a major revision of this work. In this major revision, I would expect the authors to clarify the points raised above and, most importantly, to either provide theoretical guarantees or to bring more substantial empirical evidence in favor of the claims. If the authors opt for stronger theoretical results, the following two suggestions coming from Reviewer e1Lp may be useful:

* Provide a convergence analysis involving the update interval $K$, even for very simple settings, e.g. as in https://arxiv.org/abs/1705.09886.

* Establish generalization results in simple settings, comparing them to those obtained in NTK and feature learning regimes. Ideally, one would be able to interpolate the generalization guarantees between the two regimes by varying $K$. One option is to consider the XOR problem whose population loss for a two-layer neural network is studied in https://arxiv.org/abs/2309.15111.

**Audience:**

Yes

**Audience Explanation:**

As mentioned above, the idea of iterative linearization is interesting and worth exploring. In fact, while I recommend a rejection at this stage, I would encourage the authors to submit a major revision.

**Claims And Evidence:**

No

**Claims Explanation:**

While the idea of iterative linearization is interesting, the paper fails to provide convincing evidence both at the theoretical level (little theoretical guarantees present) and at the experimental one (limited validation on practical models). The authors have provided a rather comprehensive rebuttal but these weaknesses have not been addressed in a satisfactory way. Although the recommendations of the reviewers are different and there is no consensus, all three reviewers agree that these are significant weaknesses. I concur and this is the main reason for the final decision.

**Resubmission Of Major Revision:**

The authors may consider submitting a major revision at a later time.